# Feasibility of a 12-Month Follow-Up in Swiss Older Adults after Post-Acute Care in Nursing Homes—A Pilot Study

**DOI:** 10.3390/geriatrics8020035

**Published:** 2023-03-06

**Authors:** Michael Gagesch, Andreas Hüni, Heike Geschwindner, Lauren A. Abderhalden, Wei Lang, Gaby Bieri-Brüning, Heike A. Bischoff-Ferrari

**Affiliations:** 1Department of Aging Medicine, University Hospital Zurich, 8091 Zurich, Switzerland; 2Center on Aging and Mobility, University Hospital Zurich and University of Zurich, 8006 Zurich, Switzerland; 3Geriatric Service and Nursing Homes of the City of Zurich, 8050 Zurich, Switzerland; 4University Clinic for Aging Medicine, Zurich City Hospital-Waid, 8037 Zurich, Switzerland

**Keywords:** frail elderly, feasibility study, study retention, cognitive function

## Abstract

(1) Background: Post-acute care (PAC) aims to support functional recovery in older adults after acute hospitalization in order to regain a sufficient level of self-care facilitating their return home. However, the long-term outcomes of PAC are understudied due to challenges in recording a follow-up. We aimed to investigate the feasibility of a 12-month follow-up after PAC in Swiss nursing homes, examining practicability and potential factors influencing participation rate. (2) Methods: Collection of one-year follow-up data among 140 eligible patients after PAC in nursing homes was attempted. Patients were recruited using letters and phone calls between August and December 2017. We compared baseline data of all initial PAC patients with those who declined participation in the follow-up to identify factors potentially influencing participation. (3) Results: Overall mortality at 12 months was 25% (*n* = 35 of 140). Of the 105 survivors, 53 (50%) refused participation, 26 (25%) were interviewed, and 26 (25%) were lost to follow-up. Comparison of baseline characteristics between participants and objectors indicated significant statistical differences in Mini-Mental State Examination (MMSE) scores (participants mean of 26.0 [SD 3.92] vs. objectors mean of 23.5 points [SD 4.40], *p* = 0.015). Further, logistic regression showed statistically significantly greater odds of participation (OR 1.25 [95% CI 1.06–1.48]) for each point increase in MMSE scores. (4) Conclusions: Long-term follow-up studies in older adults after PAC are challenging due to high mortality and dropout rates. Of note, intact cognitive function at baseline was associated with a higher willingness to participate in a follow-up phone interview. The assessment of cognitive function should be considered when estimating the participation rate in older patients.

## 1. Introduction

Older adults have multifaceted care needs and frequently experience negative outcomes following acute care hospitalization due to their complex health problems, including functional and cognitive impairments and frailty [1,2,3]. Thus, due to an altered level of care (ALC) following acute care hospitalization, discharge to former home settings could be unfeasible [4,5]. Consequently, care teams recommend the most appropriate discharge location for each individual in order to reduce the risk of care deficits and hospital re-admission [6].

Post-acute care (PAC) aims to provide non-acute care environments for those patients with an ALC who cannot be discharged home but at the same time are not candidates for common rehabilitation programs. This could be due to inadequate rehabilitation goals or presence of significant persisting impairments (e.g., limited weight bearing after orthopedic surgery, multimorbidity with low resilience or impaired cardiopulmonary function) [7]. Despite not being candidates for traditional rehabilitation programs, significant improvements in functional status over time can still be expected, given appropriate geriatric care and personalized therapeutic interventions [8,9].

Prior studies indicate that PAC after hospital discharge may be effective in improving physical function [9], reducing short-term mortality [10], and lowering overall costs among older adults [11]. Significant improvements in functional measures after PAC with a median duration of 31.7 days have been reported earlier by our group [9]. However, follow-up investigations regarding long-term outcomes after PAC can be challenging in this population due to difficulties in recruitment and retention as well as high mortality rates [12,13,14,15,16]. Frailty can also be regarded as an important factor influencing survival in this population [17]. In addition to the factors related to study design and execution, cognitive impairment, comorbidities, and physical frailty likely play a role in the willingness of older adults to participate in a clinical study [16].

In Switzerland, little to no robust data exists about the long-term outcomes of PAC at nursing homes, including information on independence in activities of daily living (ADL), frailty status, quality of life, and neuropsychological symptoms. The aim of this research was to address this research gap and investigate the feasibility of a one-year follow-up among 140 former PAC patients of 3 Swiss nursing homes using phone interviews. Further, a post hoc analysis was conducted to identify potential factors for effective study retention in this patient group.

## 2. Materials and Methods

### 2.1. Study Design

We conducted a 12-month follow-up pilot study to investigate the feasibility of obtaining follow-up information from a cohort of former PAC patients between August and December 2017. Subsequent to establishing informed consent, participants were interviewed using a telephone. The primary aim of our study was to describe the functional status of former patients 12 months after PAC and to assess feasibility as measured using the recruitment rate. The secondary post hoc aim was to identify potential factors associated with willingness to participate in the follow-up, including age, living status, functional independence, frailty, mental status, and cognitive function at baseline. The ethics committee of the Canton of Zurich (BASEC 2016-01069) approved our study.

### 2.2. Setting and Participants

Community-dwelling older adults aged 70 years and older after discharge from acute hospitalization who underwent PAC were eligible. Patients were obtained from three large nursing homes (a total of approximately 1600 beds of which 87 beds are reserved for PAC) in 4 specialized PAC units [18] between August and October 2016.

The PAC programs’ treatment plans were based on a comprehensive geriatric assessment upon admission. Weekly interdisciplinary team meetings supervised by a board-certified geriatrician were held to evaluate each patient’s progress and to outline treatment adjustments. The PAC program consisted of physical therapy five times per week, occupational therapy as needed, and proactive goal-oriented nursing care to support autonomy in ADL. The mean length of stay at PAC units was 31.7 (SD 16.2) days, as reported in an earlier study [9].

For the present follow-up pilot study, we contacted former PAC patients using telephone interviews 12 ± 1 months after discharge. Subsequent to identifying all deceased participants utilizing the municipal death registry, invitation letters were sent out to all survivors including information about the follow-up study as well as a return form with the option to provide or decline informed consent. In the case where an eligible patient was not able or willing to participate in the interview themselves, they were given the option to provide the phone number of a relative, trusted person, or another care provider (e.g., if living in a nursing home). Moreover, we informed former patients about an additional phone call after two weeks in the case of no written response, to provide further information, answer frequent questions, and ask for oral consent or objection. Additionally, efforts were made to reach every individual on seven different weekdays and at different times of the day to maximize the chance of enrolment.

### 2.3. Baseline Data

Baseline data from the initial PAC cohort study [9] was utilized to compare the characteristics of consenting participants with those who declined participation in the follow-up study. The extracted data were generated using assessments and interviews completed by the PAC units’ staff nurses and physical therapists upon admission to the nursing home. The data collected consisted of demographic information (age, living status, sex) and a comprehensive geriatric assessment. The assessment included a cognitive screening utilizing the Mini-Mental State Examination (MMSE), a well-validated screening tool ranging from zero to 30 points, with <24 points indicating clinically relevant cognitive impairment [19,20]. Activities of daily living (ADL) were measured using the Barthel-Index which objectively scales the patient’s functional independence and ongoing care needs from 0 to 100 points, with higher scores indicating better functional performance [21]. Frailty was operationalized with the Fried Frailty Phenotype which uses five components (unintentional weight loss, self-perceived fatigue, low physical activity, weakness, and slow walking speed) to determine frailty status (robust: 0 points, pre-frail: 1–2 points, frail: ≥3 points) [22].

### 2.4. Telephone Interview

Two senior medical students conducted the interviews using an established standardized manual to achieve high inter-rater reliability. Interviewers were instructed and trained by experienced study personnel at the Center of Aging and Mobility (the University of Zurich and University Hospital Zurich). Assessors followed a standardized manual and were trained to ask questions identically and always provide the same scripted explanations and examples.

Obtained data were entered into a standardized paper case-report form, consisting of information on living situation, caregiver role, need for formal and informal help, history of falls since discharge, and the geriatric assessment instruments mentioned below.

Since follow-up was undertaken using a phone instead of in-person assessments, the choice of instruments needed to be adapted from the initial selection to fit the telephone format. For quantifying ADL, the Katz Score consisting of 6 binary questions regarding independence in nutrition, bathing, getting dressed, going to the toilet, transfers, and continence was used [23]. Frailty was assessed using the SHARE Frailty Instrument, a questionnaire-based operationalization of the Fried Frailty Phenotype [24].

### 2.5. Statistical Analysis

To assess the recruitment rate, the frequency and percentage of participation in the 12-month follow-up assessment were calculated. Baseline demographic and clinical characteristics were determined for patients who were deceased, not reached, interviewed, and declined to participate in the follow-up telephone interview. Differences in characteristics between individuals who were interviewed and those who declined participation were tested using the Chi-square and two-sample t-test for categorical and continuous variables, respectively. A logistic regression model was fit to examine associations between demographic and clinical characteristics including age, sex, living arrangement, independence in ADL (Barthel-Index), cognitive status (MMSE), prevalent frailty (Fried-Phenotype), and willingness to participate. For frailty status, we categorized participants into frail and pre-frail/robust due to the low frequency of robust persons. All statistical analyses were performed using R Studio (R Studio, Boston, MA, USA) and SAS v9.4 (SAS Institute, Cary, NC, USA) with *p* < 0.05 considered statistically significant.

## 3. Results

### 3.1. Participants

At the 12-month follow-up, *n* = 35 (25.0%) of the initial 140 patients undergoing PAC were deceased. Invitations for follow-up interviews were sent to the remaining 105 alive patients. Of those, 23 (16.4%) declined further participation. We tried to contact the remaining 82 patients, of whom 15 (18.5%) could not be reached after multiple attempts and were consequently considered lost to follow-up, reducing our remaining sample to 56 candidates for follow-up telephone interviews. Thirty (53.5%) refused participation. In summary, 26 participants (18.5% of the initial population or 24.8% of the surviving individuals) were interviewed, either directly (65.4%) or using a proxy (34.6%). The overall drop-out rate was 79 (75.2%). A flowchart of study participation is shown in Figure 1.

### 3.2. Participation Factors

Overall, the mean age at baseline was 84.1 (±8.6) years with 62.9% (*n* = 88) females. The mean MMSE score at baseline was 24 (SD 4.5) points. A total of 77 (55%) patients were frail according to the Fried phenotype definition, 52 (37.1%) were pre-frail and 6 (4.3%) were robust. The mean Barthel-Index score at baseline was 63.2 (SD 20.0) points. Comparing baseline demographic and clinical characteristics, we found no statistically significant differences between participants who declined follow-up and participants who consented with the exception of significantly higher MMSE scores in the latter group (23.5 (SD 4.4) vs. 26 (SD 3.9) points, *p* = 0.015). The baseline demographic and clinical characteristics are summarized in Table 1.

Among the investigated covariates in the logistic regression model, only cognitive performance was statistically significant, with each higher point of the MMSE score being associated with higher odds of participation in both the univariate (OR 1.18; 95% CI 1.03, 1.37) and multivariate analyses (OR 1.25; 95% CI 1.06, 1.48). In contrast, age, sex, frailty status, ADL score, and living status were not associated with the odds of participation. The results are summarized in Table 2.

## 4. Discussion

Our pilot study identified important challenges in conducting an investigation among Swiss older adults following PAC in nursing homes, including a high mortality rate and difficulties in obtaining follow-up data. Of note, the higher-than-expected dropout rate led to an unexpectedly low number of available participants, restricting further statistical analyses. Nevertheless, we were able to identify a statistically significant association between better cognitive function and the odds of participation, with a 25% increased odds per point increment in the MMSE score. Participants who declined had a mean MMSE score of 23.5 (±4.4), i.e., below the threshold of 24 points indicative of relevant cognitive impairment. Our findings are in line with prior studies demonstrating low cognitive function as an obstacle for patient recruitment in older adults [14,25].

Regarding feasibility, we identified three main factors influencing the collection of follow-up data. First, the older Swiss adults following PAC in our study had a high 1-year mortality rate (25%) compared to the general Swiss population at age 85 of approximately 10% per year [26,27]. Nevertheless, a high mortality rate among this population might be inevitable, as older adults following PAC are often vulnerable, recently hospitalized, and frail. Despite the lack of similar studies on this specific population, our findings are consistent with earlier studies among geriatric inpatients, e.g., a 1-year mortality rate of 26.1% in older US patients discharged to skilled nursing facilities [10,12,13,28]. Therefore, in order to minimize the effect of mortality on the reduction in sample size, it is important to ensure the initial pool of eligible participants is large enough to carry out the desired analyses. Second, we were challenged with an unexpectedly high number of patients lost to follow-up. Specifically, 18.5% of the surviving patients could not be reached using their contact details. We attributed this largely to unknown fluctuations in place of residence, readmission to acute care, and subsequent permanent institutionalization. In a study by Ritt et. al, it was found that involving participants’ general practitioners (GPs) and family caregivers in study procedures appeared to effectively decrease the number of patients lost to follow-up [12]. Third, only one-third of the reached patients were willing to further participate in the follow-up. This was also an unexpected finding, although recruitment of the oldest old is known to be difficult [15,29]. A review by Forsat et al. including 50 studies identified death, withdrawal without explanation, and health problems as the three main barriers impeding the retention of older adults in clinical trials [30]. The authors also conclude that the information about reasons for study discontinuation in older adults is still limited [30]. Our study was not designed nor intended to collect sufficient information to identify those factors.

## 5. Strengths and Limitations

This study has several strengths. To our knowledge, this is the first attempt at a follow-up study in Swiss older adults after PAC in a nursing home setting. We conducted interviews following a strict protocol including standardized explanations and scripted answers to avoid discrepancies and minimize interviewer bias. Further, the initial set of clinical assessment tools and questionnaires consisted of well-validated instruments. In addition, our study population consisted of a patient group, largely underrepresented in the majority of geriatric research.

Our study also has limitations. Baseline data collection in the original sample took place under real-world conditions and patients might not have recognized their consent to “further use” of their regularly obtained patient data as being eligible candidates for future studies. In that, patients who were contacted for this study might not have associated this inquiry with their prior PAC stay. In addition, the interviewers were strangers to the patients and a distrust of research assistants is a known obstacle for studies in older adults [16]. In earlier studies and reviews concerning the recruitment and retention of older adults in research, study awareness as well as personal contact were pointed out to be effective strategies for increasing the recruitment rate [25,30,31]. In addition, we did not have information on the educational attainment of our participants. While we focused on the specific group of mostly pre-frail and frail former PAC patients, findings might not be applicable to the heterogeneous group of community-dwelling older adults. Another limitation is the restricted timeline of this study. Our study was planned post hoc after the completion of baseline data collection. Thus, patients were not aware of this study opportunity at the time of initial data acquisition. This could have created possible bias in terms of recruitment success and thus feasibility. Finally, our investigation was limited to a pilot study assessing study feasibility to be used in a larger study and was not intended to gain broad insights from this population.

In summary, our follow-up study among former PAC patients faced several barriers in terms of recruitment and was finally determined unfeasible. Strategies to raise the participation rate should include raising awareness, a personalized approach, and the involvement of the patient’s GP or proxies. There could be several strategies to increase participation in future studies considering these. One could be the pre-emptive involvement of GPs and family members. In addition to an expected decrease due to loss from follow-up, they can contribute by motivating patients on a more personal level [32]. A more time-consuming approach could be recruitment or even assessment in person by the research team [33]. Additionally, interim phone calls could be conducted, e.g., 1, 3, and 6 months after recruitment, as a reminder and possibility to gather additional information; this approach creates trust in addition to awareness [34].

## 6. Conclusions

Conducting successful follow-up studies in older adults after PAC is challenging. In addition to a sufficiently large sample of eligible patients at the study start, proactive and comprehensive involvement of caregivers, GPs, and care services during follow-up could be key factors. In addition, an impaired cognitive function might be considered as a potential barrier to study retention in this population.

## Figures and Tables

**Figure 1 geriatrics-08-00035-f001:**
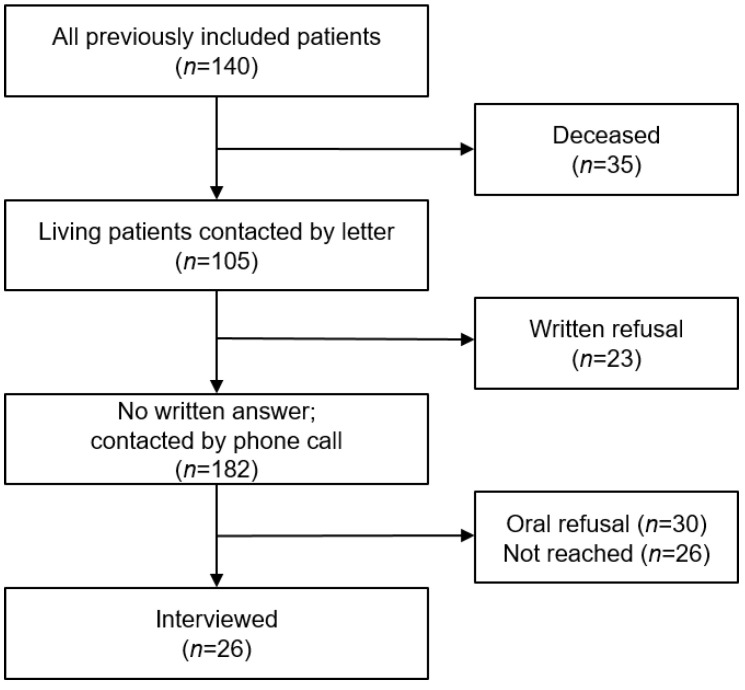
Participation flow chart.

**Table 1 geriatrics-08-00035-t001:** Baseline characteristics.

	Overall	Deceased	Not Reached	Interviewed	Declined	*p* Value ^a^
	(*n* = 140)	(*n* = 35)	(*n* = 26)	(*n* = 26)	(*n* = 53)	
Age, mean (SD)	84.1 (8.57)	87.3 (7.63)	81.8 (9.34)	85.5 (7.60)	82.5 (8.66)	*0.13*
Female, *n* (%)	88 (37.1%)	19 (54.3%)	13 (50.0%)	20 (76.9%)	36 (67.9%)	*0.57*
Living Status, *n* (%)						
alone	87 (62.1%)	16 (45.7%)	21 (80.8%)	16 (61.5%)	34 (64.2%)	*0.17*
family/friend	8 (5.7%)	3 (8.6%)	1 (3.8%)	3 (11.5%)	1 (1.9%)	
spouse	44 (31.4%)	15 (42.9%)	4 (15.4%)	7 (26.9%)	18 (34.0%)	
MMSE, mean (SD) ^b^	24.0 (4.46)	23.6 (4.54)	23.8 (4.73)	26.0 (3.92)	23.5 (4.40)	** *0.015* **
Frailty Status ^b^						
robust	6 (4.3%)	0 (0%)	3 (11.5%)	0 (0%)	3 (5.7%)	*0.33*
pre-frail	52 (37.1%)	6 (17%)	13 (50.0%)	10 (38.5%)	23 (43.4%)	
frail	77 (55.0%)	27 (77.1%)	9 (34.6%)	16 (61.5%)	25 (47.2%)	
Barthel-Index, mean (SD)	63.2 (20.0)	63.2 (20.0)	66.7 (22.7)	61.9 (20.6)	64.9 (18.6)	*0.52*

^a^ testing for differences between interviewed and declined; ^b^
*n* = 2 missing values in MMSE and frailty status for the declined group.

**Table 2 geriatrics-08-00035-t002:** Odds of participation in follow-up.

	Univariate Model (*n* = 79)	Multivariate Model ^a^ (*n* = 77 ^b^)
	OR (95% CI)	OR (95% CI)
Age	1.05 (0.98, 1.12)	1.06 (0.98, 1.14)
Men (Women as reference)	0.64 (0.22, 1.87)	0.66 (0.18, 2.42)
Living Situation Home alone (Reference)	1.0	1.0
Family/Friend	6.38 (0.61, 66.17)	9.74 (0.49, 194.56)
Spouse	0.83 (0.29, 2.38)	1.39 (0.38, 5.1)
MMSE (one point increase)	**1.18** (1.03, 1.37)	**1.25** (1.06, 1.48)
Frailty Status ^c^ (robust/pre-frail as ref.)	1.0	1.0
Frail	1.67 (0.64, 4.36)	1.17 (0.39, 3.55)
Barthel-Index (one point increase)	0.99 (0.97, 1.02)	0.99 (0.96, 1.02)

^a^ Adjustment variables: age, sex, living situation, MMSE, frailty status, and Barthel-Index. ^b^ Due to two NAs in the frailty status, those observations are excluded and, therefore, the sample size drops to 77. ^c^ Frailty status of pre-frail and robust were combined.

## Data Availability

Data are available upon justified request to the authors.

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
