# Peer review of "Feasibility of a 12-Month Follow-Up in Swiss Older Adults after Post-Acute Care in Nursing Homes—A Pilot Study"

_geriatrics, 2023, doi:10.3390/geriatrics8020035_

Round 1

Reviewer 1 Report

Dear Authors, 

Thank you for the opportunity to review manuscript entitled “Feasibility of a 12 months follow-up in Swiss older adults after 2 post-acute care in nursing homes - A pilot study". 

I think the manuscript raises important issues, but some corrections are needed.

Introduction

·       Please specify the purpose of the study, was the aim of the study an assessment of factors for continuing the study? In the discussion, the authors state that the study was not designed to identify reasons for study discontinuation in the older people.

Materials and methods

·       how was the sample size calculated? Are 26 observations sufficient for preliminary conclusions from the pilot study?

Results

·       in the tools, the authors provide more scales and tests used in the study. Please add other results to the group description: ADL; IADL, SHARE Frailty Instrument, EuroQoL, VAS and Mental status (PHQ-2).

·       Why none of these results is not in the tables, and why MMSE and Index Barthel appear?

Summary

I think the study is interesting, but please specify the purpose of the analysis, describe the study group using appropriate tools and use these data obtained in the analysis.

Currently, I don't really understand why other tools appear in the analysis.

Reviewer 2 Report

Dear authors,

Thank you for the opportunity to review your paper. The topic of your paper is relevant and of growing interest in an aging population.

There are some minor remarks:

L32 com-pared
L40 bea-ring etc.

manuscript is full of typos like that. I suggest a native speaker to your manuscript.

Major concerns:

Please justify sample size of this study due to high incidence of dropout and mortality. Was there any sample size calculation? As you state that drop out was much higher than expected what rate did you assume prior to the study?

Please justify your approach as compliance seemed to be a major problem. You discussion should take into account that other approaches might have been tested. Why didn´t you chose another one?

I don´t understand the objective of your study. What was your primary endpoint? Willingness to participate in a survey? Please clarify your main objective.

Another limitation is that data now is more than 5 years old as the study was conducted way before the COVID19 pandemic. What might have changed afterwards in your target group (e.g. openess to research staff, increased overall digital literacy making it possible to use different approches).

Conclusion:

I would not agree on the need of cognitive screening to improve participation rate, but to seek for different approaches in order to ensure higher participation. Also the second sentence seems to generic for me.

In my view this study is underpowered. Additional participants would have been needed to draw clear conclusions.

Reviewer 3 Report

The manuscript describes well the research question, methodology, results and links to the current literature and gaps.

With regards to the study design (and limitations), the manuscript could benefit from addressing other determinants of cognitive impairment, e.g. the level of education (as described in Zahodne et al (2015). Differing effects of education on cognitive decline in diverse elders with low versus high educational attainment. Neuropsychology, 29(4), 649: "More years of education was associated with higher cognitive level and slower cognitive decline"). 

Author Response

Author’s reply to reviewer 3

 Comment 1

 With regards to the study design (and limitations), the manuscript could benefit from addressing other determinants of cognitive impairment, e.g. the level of education (as described in Zahodne et al (2015). Differing effects of education on cognitive decline in diverse elders with low versus high educational attainment. Neuropsychology,  29(4), 649: "More years of education was associated with higher cognitive level and slower cognitive decline").

 Response to comment 1

 We thank the reviewer for this statement, to which we agree. Unfortunately, we do not have the information on the years of education of our study participants and added this information to the limitations.

Round 2

Reviewer 2 Report

Dear authors,

thank you very much for your revision of the manuscript, also justifing estimated (not calculated) sample size.As you mentioned that your pilot study does not fit to test on hypothesis I am still wondering how you were able to calculated and detected siginifant correlation.

I am fine with most other answers. However limitations as discussed in your response (e.g. new post pandemic approaches concerning digital technologies to enhance compliance) may be also considered to enrich the discussion part of your paper. I would leave decision on this to the discretion of the editor.

Good luck!